

**Polycyclic aromatic hydrocarbon in urban soils of the Eastern European megalopolis: distribution, source**
**identification and cancer risk evaluation**
**Shamilishvily George Avtandilovich[1]** (corresponding author), e-*mail: george199207@gmail.com, +79217868236*
**Abakumov Evgenii Vasil'evich[1]**, *e-mail: e_abakumov@mail.ru*
**Gabov Dmitrii Nikolaevich[2]**, *e-mail: gabov@ib.komisc.ru*
[1]*St. Petersburg State University, Dept. of Applied Ecology, St. Petersburg, Russia*
[2]*Komi Biological Institute of the Russian Academy of Sciences, Syktyvkar, Russia*
**Abstract**
The study explores qualitative and quantitative composition of 15 priority PAHs in urban soils of some
parkland, residential and industrial areas of the large industrial center in the Eastern Europe on example of Saint-
Petersburg (Russian Federation). Aim of the study was to test the hypothesis on the PAH loading differences
between urban territories with different land use scenarios. Qualitative and quantitative determination of PAHs in
soils was carried out by reverse-phase high-performance liquid chromatography (HPLC). Benzo(a)pyrene toxic
equivalency factors (TEFs) were used to estimate benzo(a)pyrene equivalent ($BaP_{eq}$) concentrations in order to
evaluate carcinogenic risk of soil contamination with PAHs. Results of the study demonstrated that soils within
urban areas with different land utilization types are characterized by common loads of PAHs generally attributed to
high traffic activity in the city. Considerable levels of soil contamination with PAHs were noted. Total PAH
concentrations ranged from 0.33 to 8.10 mg·kg$^{-1}$ and showed no significant differences between land utilization
types. The common tendency in PAH distribution patterns between investigated sites clearly indicates the common
source of PAHs in urban soils. A larger portion of high molecular weight PAHs along with determined molecular
ratios suggest the predominance of pyrogenic sources, mainly attributed to combustion of gasoline, diesel and oil.
Petrogenic sources of PAHs have a significant portion as well defining the predominance of petroleum associated
low molecular weight PAHs such as phenanthrene. Derived concentrations of 7 carcinogenic PAHs as well as
calculated BaP total potency equivalents were multiple times higher than reported in a number of other studies,
indicating a significant risk for human health in case of direct contact. The obtained $BaP_{eq}$ concentrations of the sum
of 15 PAHs ranged from 0.05 to 1.39 mg·kg$^{-1}$. A vast majority of examined samples showed concentrations above
the safe value of 0.6 mg·kg$^{-1}$ (CCME, 2010). One-way ANOVA results showed significant differences in levels of
pyrene, fluoranthene and phenanthrene – the most abundant individual PAHs in examined sampled, between
parkland, residential and industrial land uses, suggesting the influence of land use factor on distribution of these
pollutants. Exposure to these soils through direct contact probably poses a significant risk to human health from
carcinogenic effects of PAHs, even in urban parklands.
**1. Introduction**
The quantity of toxic organic substances is extremely high, but in the world practice the evaluation of
contamination levels of certain areas is produced mostly for polycyclic aromatic hydrocarbons (PAHs), an
ubiquitous organic pollutants in environments, particularly in soils and sediments. PAHs are a large group of
persistent organic compounds (POPs) consisting of several hundred individual homologues and isomers containing
at least two condensed aromatic rings. Their input to the environment has both natural and anthropogenic origins.
Natural sources includes releases from vegetation fires, diagenetic processes and volcanic exhalations (ATSDR
1995; Wilcke 2000). In turn, anthropogenic PAHs occur from pyrolytic processes, especially incomplete combustion
of organic during industrial activities, domestic heating, waste incineration, transportation and power generation
(ATSDR 1995; Wilcke 2000; Dyke et al. 2003). It is believed that by far most PAHs are released into environment
by anthropogenic combustion of wood and fossil fuels (Wilcke 2000). Sign of anthropogenic contamination of soil
with PAHs are even detected in such remote places as Antarctic Stations, which origin is doubted, whether it has
natural sources, i.e. decomposition of plant and guano materials, or comes from anthropogenic sources, such as fuel
combustion, petroleum products and long range transport with atmospheric solid particles (Abakumov et al. 2014;
Abakumov et al. 2015). Some PAHs are of the most environmental importance because of the established
carcinogenic, mutagenic and teratogenic effects in living organisms and in humans particularly (Menzie et al. 1992;
Shaw and Connell 1994; Perera 1997; Yu 2002; Guo et al. 2013). A number of 16 PAHs have been listed as priority
contaminants by both the US Environment Protection Agency (US EPA) and European Union (EU). Among them
seven compounds, i.e. benzo(a)anthracene, chrysene, benzo(a)pyrene, benzo(b)fluoranthene, benzo(k)fluoranthene,
dibenz(a,h)anthracene and indeno(1,2,3-cd)pyrene are considered as probable human carcinogens (US EPA 2002).
In Canada, US and some European countries normalization of soil contamination is provided upon developed soil
quality criteria for selected PAHs or their sum. Only a few countries have established comprehensive soil guideline





values (SGV) for particular land use at least for the sum 85 of priority PAHs (Σ7; 10; 15; 16). Generally, the existing soil critical values provides only human health-risk based approaches and don't consider protection of other ecological receptors. In turn, US EPA has developed ecological soil screening levels (Eco-SSLs) for PAHs, which are derived separately for four groups of ecological receptors: plants, soil invertebrates, birds and animals. However these screening levels are intended to evaluate an unacceptable ecological risk to terrestrial receptors, they are not designed to be used as cleanup levels. For this purpose US EPA adopted a human health based Preliminary Remediation Goals for Soil (PRG) using estimates of different routes of exposure. In contrast to this, Russian Federation has not yet developed soil guideline values at least for the sum of priority PAHs; normalization is provided only for soil contamination with benzo(a)pyrene without distinction for particular land use. On top of that, no threshold values are provided for other POPs. A summary of soil guideline values for PAHs set in some countries is presented in **supplementary materials 1.** Thus studies on soil contamination with PAHs are of the most importance as they provide information that can be further used to delineate special contaminated sites exhibiting high risk to human exposure. Thousands of reports about PAHs concentrations, sources and health risk assessments in urban and semiurban areas from all over the world were published in recent years. Elevated levels of PAHs in urban soils were reported in Hustan, USA (Hwang et al. 2002), Beijing, China (Tang et al. 2005), Glasgow of UK and Torino of Italy (Morillo et al. 2007), and Esbjerg of Denmark (Essumang et al. 2011).

St. Petersburg is the largest industrial and transport center in the northwestern region of Russia and is of great interest from the viewpoint of environmental concern. The ecological status of such a large center reflects the whole range of socioeconomic problems resulting in decline of human health under the influence of various chemical, physical and biological factors. The ecological situation in the city is determined by the emissions from more than a thousand industrial enterprises, large railway junction, seaport and the great motor vehicle fleet – 1 670 794 cars and 207 975 trucks as of 2014 (Belousova et al. 2014). All this transport is served by a huge amount of petrol stations and transport companies: currently in St. Petersburg operate 27 fuel operators and 397 petrol stations. Industrial enterprises of the city include high-capacity, resource- and power-consuming ecologically dangerous works. According to the data collected from the automatic air monitoring system of the city in 2014 total emission into the air from both the stationary sources and vehicles has reached 513 200 t in 2014 of chemicals, including 16 903 t of hydrocarbons (CHx), 3000 t of black carbon (BC) and 47 900 t of volatile organic compounds (VOC) (Belousova et al. 2014). The amount of emissions per capita complies 135.9 kg / year, per unit area – 434.5 t / km2 (Belousova et al. 2014). At the same time, 91.9% of emissions are accounted to the transport activity. Industrial and transport emissions are the major source of soil contamination with PAHs in urban areas along with inputs from petroleum products. No systematic survey of soil contamination with priority PAHs has been conducted yet in St. Petersburg except for benzo(a)pyrene (Gorkiy and Petrova 2007). Considering this fact and environmental aspects of the territory described above, St. Petersburg affords an excellent location to study geochemical cycles of PAHs.

Therefore this study is aimed to test the hypothesis on the PAH loading differences between urban territories with different land use scenarios. The results of this study would contribute to the knowledge about PAH distribution in urban soils of Eastern European region and may be used by decision makers during land management.

Objectives of the study were to: 1) explore qualitative and quantitative composition of 15 priority PAHs in urban soils in some parkland, residential and industrial areas of St. Petersburg: 2) compare with existing data on the PAHs distribution in urban soils; 3) distinguish between PAHs sources using PAH molecular ratios; and 4) evaluate cancer risks associated with soil contamination with PAHs within selected areas.

## 2. Materials and methods
### 2.1 Study site description

Choice of the study area, namely Primorsky, Vasileostrovsky and Kirovsky administrative Districts of St. Petersbrg was done in order of increasing of location density of potential stationary sources of contamination with PAHs, population density and traffic activity. Detailed characteristics about each chosen area are given in Table 1. Certain areas of recreational, residential and industrial land use scenarios within each chosen District were subjected under the study. Information on the land use scenario of each chosen area was obtained using online map service "Regional Geoinformational System RGIS" developed with the support of the Committee for land resources and land management of St. Petersburg (Fig. 1). Potential sources of PAH contamination affecting PAH levels in soil here are high traffic activity (Western highway and Primorsky prospect), steel and chemical industries (Kirovsky engineering plant, Baltiysky shipyard plant, varnish factory "Kronos"), thermal-power-stations ("North-Western").



Climate is moderately continental, significantly affected by the Baltic Sea. The average annual amount of precipitates varies 565-635 mm. Humidity factor – 1.1-1.3. The territory represents an almost flat plain with altitudes below 20 m above the sea level (Neva Lowland). Natural soil formation usually occurs on ancient lake-marine littoral sands, sandy loams, loams (less) depleted in calcium (Gagarina et al. 2008). Urban soils are formed on the bulk deposits ranging from 0.9 to 4 m of thickness (Matinyan et al. 2005; Dashko et al. 2011). Soils are strongly disturbed by anthropogenic activities (buried, sealed and/or contaminated), with small relatively intact islands in natural and seminatural areas on the north, north-west and north-east of the City. Soils of the historical center are presented by anthropogenic soil-like formations called in national soil classification systems as "*urbanozems*" (Stroganova e. al. 1992) or "*urbiquazizems*" (Shishov et al. 2006) and generally characterized by light grain size, modified soil profiles, with abundant inclusions of anthropogenic artefacts in the form of debris, domestic wastes and remains of communications, neutral to alkaline pH, high humus, nitrogen and phosphorus content, humate and fulvic-humate types of humus and traces of chemical contamination (Rusakov et al. 2005; Matinyan et al. 2005; Ufimtseva et al. 2011). Investigated urban soils were classified as *Technosols* according to the World Reference Base for soil resources (Michéli et al. 2006).

### 2.2 Sampling strategy and procedure

Sampling was conducted in September 2013 at 9 urban sites, in dry and clear weather conditions according to international standard protocol ISO 10381-1 (2002) and national sampling standard GOST 17.4.4.02-84 (1984). Soil samples were taken from 0-20 см topsoil layer. A total of 135 grab soil samples were collected diagonally from 25 $m^2$ sampling plots were combined into 27 composite samples of 0.7 kg each one. Location of the sampling sites was defined according to proximity to residential areas and potential pollution sources (Fig.1).

Sampling strategy responds to the study objectives and is aimed to provide comprehensive characterization of the selected sites suspected to be contaminated with PAHs. Quantity of grab samples to be collected depended on the size of sampling sites, e.g. 15 grab samples per 0.8 ha site collected from sampling plots (S = 25 $m^2$). Soil depth selected for sampling is a function of exposure routes (e.g. soil ingestion, dermal contact with soil and dust, inhalation of contaminated dust, inhalation of volatile compounds). Sampling pattern represents both the purposive and judgement sampling techniques, delineating sample locations that assumed to be representative of the whole site and most contaminated. Instruments for sample derivation included stainless scoop and stainless knife prewashed with acetone. The representativeness of collected samples was provided thorough mixing and taking an average sample by quartering method.

**Fig. 1.** Location of the soil sampling sites.

| Description to fig. 1 | | | |
|---|---|---|---|
| Land use | **a** - Primorskij District | **b** - Vasileostrovskij District | **c** - Kirovskij District |
| Parkland | **1** - The park of the 300th anniversary of St. Petersburg, Primorskij prospect, 157. 59°59′2″ N, 30°11′33″ E. | **4** - AkademicheskijGarden, 2d Line of Vasilyevskij Island, 2A. 59°56'19.8" N, 30°17'18.3" E. Rumyantsevskij Garden, Rumyantsevskaya square, 7. 59°56'18.4" N, 30°17'33.1" E. | **7** - The park of 9th January, Stachek prospect, 19. 59°53'31.1" N, 30°16'25.5" E. |
| Residential | **2** - Intersection of Yahtennaya street and Optikov street. 59°59'55.7" N, 30°13'22.9" E. | **5** - Korablestroiteley street, 20, 19/2. 59°56'37.3" N, 30°12'48.3" E. 59°56'38.0" N, 30°13'05.2" E. | **8** - Korneev street, 4. 59°53'06.9" N, 30°16'03.8" E. |
| Industrial | **3** - Vicinity of the Bus depot №2, Avtobusnaya street, 12A. 60°01'46.6" N, 30°15'34.7" E. | **6** - Vicinity of the Baltic shipyard, Detskaya street, 3. 59°55'36.1" N, 30°15'13.1" E. | **9** - Vicinity of the Kirovskij engineering plant, Stachek prospect, 47. 59°53'09.3" N, 30°15'48.1" E. |

Collected samples were packed in labeled sterile plastic bags, kept in cool condition and transported to the laboratory. Once in laboratory, soil samples were dispersed on the sterile glass plates and air-dried at the room temperature for 5 days, cleaned from the organic and inorganic debris, grounded in laboratory vibrating cup mill, sieved through 0.25 mm caprone sieve and finally stored in the dark glass containers prewashed with acetone until analysis. This technique enables to prevent cross-contamination as well as losses of PAHs due to environmental factors (Berset et al. 1999).





**2.3 HPLC, PAH source identification and risk evaluation**
15 PAHs were analyzed, including naphthalene (NAP), acenaphthene (ANA), fluorene (FLU),
phenanthrene (PHE), anthracene (ANT), fluoranthene (FLT), pyrene (PYR), benzo(a)anthracene (BaA),
chrysene (CHR), benzo(b)fluoranthene (BbF), benzo(k)fluoranthene (BkF), benzo(a)pyrene (BaP),
dibenz(ah)anthracene (DBA), benzo(g,h,i)perylene (BPE), indeno(1,2,3-cd)pyrene (IPY) (**Fig. 2**).
**Fig, 2.** Structures of the studied PAH compounds
PAHs content in samples were determined on the basis of US EPA method 8310 (1996a), national standard
method PND F 16.1:2:2.2:3.62-09 (2009), and Gabov (2007; 2008). Extraction of the PAHs was carried out at room
temperature with methylene chloride (high purity grade) and ultrasonic treatment via Branson 5510 ultrasonic bath
(USA, power 469 W, working frequency 42 kHz) following the US EPA method 3550b (1996b). Solvent removal
(evaporation) was carried out with Kuderna–Danish concentrator (Supelco). PAHs fractions were purified by
consecutive chromatography in columns filled with aluminum oxide (Brockman activity grade 2-3, Neva Reaktiv)
and silica gel (Fluka) according to the US EPA purification method 3660c (1996c). The purity was controlled by the
absence of peaks in the blank chromatogram. A standard mixture of 15 PAHs (Supelco) with the concentrations of
each component in the range of 100–2000 $\mu g/cm^3$ was used to prepare the standard PAH solutions. Qualitative and
quantitative determination of PAHs in soils was carried out by reverse-phase high-performance liquid
chromatography (HPLC) in gradient mode with spectrofluorimetric detection via chromatograph "Lyumahrom"
("Lumex", Russia). Chromatography was performed at 30°C on a column Supelcosil ™ LC-PAH n5 μm (25 cm ×
2.1 mm). Mobile phase was provided with acetonitrile-water gradient. Samples of 10 μl volume were injected using
injection valve. Individual PAHs were identified by the time of retention and comparison of fluorescence spectra of
the components coming from the column with spectra of the standard PAHs. Quantitative analysis of PAHs was
performed by external standard method. For the quality assurance purposes Standard reference materials® 1944
New York/New Jersey Waterway Sediment (National Institute of Standards and Technologies NIST, USA)
containing a mixture of 15 PAHs were subjected to the procedure described above. The error of measuring the
PAHs (benz[a]pyrene) in the soils was 35% in the range of 5–40 ng/g and 25% in the range of 40–2000 ng/g with a
confidence probability of P = 0.95.
PAH molecular markers and ratios were used to determine PAH sources (Pandey et al. 1999; Yunker et al.
2002; Hwang et al. 2003). Sum of combustion PAHs (CombPAH/15PAH) was used as tracer of pyrogenic sources.
CombPAH/15PAH marker indicates portion of the sum of combustion specific compounds in total PAH content,
which are Fluoranthene, Pyrene, Chrysene, Benzo(a)anthracene, Benzo(k)fluoranthene, Benzo(b)fluoranthene,
Benzo(a)pyrene, Benzo(g,h,i)perylene and Indeno(1,2,3-cd)pyrene (Prahl and Carpenter, 1983). Applied PAH
molecular markers and ratios as well as their ranges are given in **supplementary materials 2**.
Since benzo[a]pyrene (BaP) is the most studied PAH, the carcinogenic potential of other PAHs is generally
assessed referring it to that of BaP ("toxicity equivalence factors" (TEFs), in similar way to the "toxic equivalents"
(TEQ) used in the evaluation of the toxicity of dioxins and furans. Benzo[a]pyrene Potency Equivalence Approach
is a major approach used by regulatory agencies such as the US EPA (1993; 1999), California EPA (OEHHA 1992),
Netherlands (Verbruggen et al. 2001), the UK (Duggan and Strehlow 1995), or Provinces of British Columbia and
Ontario for assessing the human health risks of PAH-containing mixtures.
**2.4 Soil properties analysis and statistical treatment**
Total organic carbon (TOC) was determined using a "Leco" CHN-628 elemental analyzer (USA,
combustion temperature 1030 °C, oxygen boost time 28 s). Inorganic carbonates were removed before analysis by
acidification in situ of the grounded samples with 1 M hydrochloric acid in order to avoid uncertainty in TOC
determination. Clay content was determined with laser diffractometer "Shimadzu" SALD-2201 (Japan). All
measurements were done in triplicate. All measurements were converted to absolutely dry sample. Data on analyzed
properties of the studied soils is presented in **Table 2**.
Measured TOC concentrations in studied samples ranged between 3.82 to 6.41% with a median value of
4,80%. Numerous studies suggested that soil organic matter (SOM) content plays an important role in retention of
PAH in soil (Conte et al. 2001; Chung and Alexander 2002). In simple terms the higher SOM concentrations are,
than the higher amount of PAHs can be absorbed (Karickhof and Brown, 1979; Wilcke, 2000). Entering the soil
from the atmosphere PAHs are preferentially sorbed to aggregate surfaces (Wilcke, 1996). The close association of
PAHs with SOM results in differentiation of organic contaminants pools among particle-size fractions
(Guggenberger et al. 1996). A significant increase of PAH concentrations in finer fractions is shown in a number of



studies (Wilcke, 1996). Clay content in studied soils ranges between 1.87 and 8.50 %. A correlation coefficients
were calculated in present study in order to reveal relationship between levels of PAH in soil and analyzed soil
parameters. A strong positive correlation was found between sum of 15 PAH in soil and clay content (r = 0.91; n =
27; p = 0.95), however, no correlation of total PAH and TOC concentrations in soil was detected.
Statistical treatment of the data was carried out with STATISTICA 10.0 software. One-way ANOVA was
applied in order to test statistical significance of differences between obtained data. The essence of the method is
based on estimation of the significance of averages differences between three or more independent groups of data
combined by one feature (factor). The null hypothesis of the averages equality is tested during the analysis
suggesting the provisions on the equality or inequality of variances. In case of rejection of null hypothesis basic
analysis is not applicable. If the variances are equal, F-test Fisher criterion is used for evaluation of intergroup and
intragroup variability. If F-statistics exceeds the critical value, the null hypothesis is rejected considering inequality
of averages. Post-hoc-test (Fisher LSD) was used to provide detailed evaluation of averages differences between
analyzed groups of data. A feature of post-hoc-test is application of intra-group mean squares for the assessment of
any pair averages. Differences were considered to be significant at the 95% confidence level. All calculations were
carried out via STATISTICA 10.0 software. PAH concentrations were analyzed at least in triplicate. Calculated
mean concentrations were provided with standard deviations (a ± b).

### 3. Results and discussion

### 3.1 PAH concentrations in studied soils

The levels of 15 individual PAH compounds analyzed in soils are shown in **Table 3. T**he sum of 15 PAH
and the sum of 7 compounds included in the group of probable human carcinogens (B2) by the US EPA (1993) are
given additionally. Total PAH concentrations in studied soils were found to range from traces to 8.06 mg·kg$^{-1}$ (sum
of 15 priority PAH, hereafter referred to 15PAH). The vast majority of samples were characterized by
concentrations of more than 1 mg·kg$^{-1}$, which is set as a guide level for total PAH content in soil by a number of
countries. The highest 15PAH levels were observed in soil samples collected from residential and industrial sites
reaching an average of 4.19 and 4.01 mg·kg$^{-1}$ respectively with a maximal value of 8.06 mg·kg$^{-1}$ for industrial site in
Kirovsky district (hereafter – K.D.) Concentrations found in parkland areas were substantially lower than those of
residential and industrial, with an average value of 1.08 mg·kg$^{-1}$.
Distribution of the sum of the 7 carcinogenic PAH (7PAH) in soils of the studied urban sites is generally
characterized by the same pattern as the total PAH content in soils. The highest 7PAH levels were measured in soil
samples taken from residential sites (1.94 mg·kg$^{-1}$) with an absolute value of 3.47 mg·kg$^{-1}$ in technosol of K.D.
residential area. 7PAH levels in parkland areas tend to be at lower range respectively to distribution of 15PAH. All
sampling sites were located in a proximity of less than 250 m to the highways (Korablestroiteley street, Stachek
prospect, Optikov prospect, University embankment, Bolshoi prospect V.O. and others) showing heavy traffic. The
portion of 7PAH to the 15PAH in all tested samples ranged between 41 % and 46 %, which evidently shows that the
soils may represent considerable health risk for human.
The bar chart showing the contribution of PAH with different ring numbers to the sum of PAH in soils is
depicted in **Fig. 3.** The sum of organic pollutants is mostly dominated by heavy molecular weight PAH with 4-5
rings. Portion of 4-ringed PAH compounds in soil of residential and industrial sites accounts for 50% of the sum
decreasing to 34% in parkland soils. 5 ringed PAH including such compounds as BaP, BbF, BkF, and DBA
contribute up to 31 % of the sum of PAH insignificantly varying between studied areas. The rest portion is
accounted for the 6-ringed (10-14%) and low molecular weight PAHs with 2 or 3 rings in structure (11-17%).
**Fig 3.** Distribution pattern of PAHs with different ring numbers in studied soils
The pie chart illustrating composition of PAH mixtures in soils is depicted in **Fig. 4.** The obvious equality
in PAH distribution patterns in all studied sites clearly indicates the common source of PAHs. Pyrene and
Fluoranthene (4-ring PAHs) are the most abundant compounds in examined samples, portion of which accounts for
16-18 % of 15PAH. The following predominant compounds are 5-ring PAH benzo(b)fluoranthene (10-11%) and
benzo(a)pyrene (8-11%). The rest portion of the sum is represented by lighter weight PAHs (2-3-ring PAHs) and is
generally dominated by Phenanthrene (6-9%). Domination of 4 and 5-ring PAHs, mainly PYR, FLT, BbF and BaP,
in studied soils is indicative of elevated diesel fuel consumption activity on the territory. Estimated diesel
consumption in St. Petersburg reaches 38% of the total fuel use for transportation (Belousova et al. 2014). As known
emission rate of heavyweight PAH fraction due to diesel combustion is several times higher (Sjogren et al. 1996;
Marr et al. 1999).



**Fig 4.** Composition of PAH mixtures in studied soils

248       Obtained data are nearly consistent with data from Lodygin et al. (2008) exploring PAH levels (sum of 11
PAHs) in soils of Vasilievsky Island in St. Petersburg). The main anthropogenic impact on soils of residential area
of the island was exerted by light polyarens, including 2-4 ring substances (as stated by the author), the portion of
which in the total content of PAHs was more than 50%. Maximum concentrations of PAHs were detected in soils
along highways with intense traffic and considerable emission of combustion gases. The reported total PAH content
ranged from 0.197 to 8.20 mg·kg$^{-1}$ between different land utilization types. The described distribution patterns of
individual PAHs are similar to those of this study: the most abundant are 4-5 ring PAHs, particularly Pyrene (17%),
Fluoranthene (17%), Benzo(g,h,i)perylene (13%), Benzo(b)fluoranthene (12%) and Benzo(a)pyrene (12%). Several
samples were noticed to exhibit higher contents of heavy polyarens of natural origin, as both of the samples were
represented by fresh organic material (peat) which is used as amendment in soils of residential areas and roadsides.
Thus the findings of above mentioned study suggest that spatial distribution of PAHs is mainly dictated by the
closeness to highways and by the artificial input of peat material in the urban soils.

260       There is still a lack of information about PAHs concentrations in soils of St. Petersburg, so the data on the
pollutants distribution in water sediments obtained from environmental monitoring systems may be applied in
discussion for evaluation of the PAH loads. Comparative PAH levels were detected in bottom sediments in different
parts of Neva Bay (Gulf of Finland) and along the Niva river waterway. Reported total PAHs concentrations ranged
between 0.01 to 14.5 mg·kg$^{-1}$ (HELCOM 2014). Benzo(a)pyrene was detected in 96% of sediment samples taken
with and average concentration of 0.09 mg·kg$^{-1}$.

266       Total PAH concentrations in soils of urban and industrial sites from a number of investigations set in other
countries are summarized in **Table 4.** Tang et al. (2005) reported a sum of 16 PAHs of 27.82 mg·kg$^{-1}$ in roadside
soils of Beijing, China. Hwang et al. (2002) found a total PAH concentration of 0.20-2.20 mg·kg$^{-1}$ in urban and
suburban soils in Huston, Texas, USA. Notable PAH concentrations were observed by Mielke et al. (2001) in New
Orleans urban soils (USA), fluctuating around a medium level of 3.73 mg·kg$^{-1}$. Nadal et al. (2004) reported
relatively lower 16 PAHs levels in soils of the vicinity of the chemical and petrochemical industries and
urban\residential sites in Tarragona County (Catalonia, Spain), ranging 284 between 0.11 and 1.0 mg·kg$^{-1}$.
Comparable findings were announced by Bucheli et al. (2004) for soils of urban and semiurban areas in Switzerland,
containing 0.05-0.62 mg·kg$^{-1}$ of the sum of 16 PAHs. In general terms, the predominance of 3-5 ring PAHs is noted,
which is mainly attributed with influence of the anthropogenic activities on the studied territories.
**3.2 Determination of the PAH sources and statistics**

277       While a domination of high molecular weight PAH fraction indicates a combustion origin (pyrogenic),
enrichment of low molecular weight PAHs is common in fresh fuels (petrogenic) (Masclet et al. 1987, Budzinski et.
al. 1997). Special molecular markers and ratios, proposed by Yunker et al. (2002) and a total combustion PAHs
index, reported by Hwang et al. (2003) were applied for PAH sources apportionment. Obtained meanings of applied
PAH molecular ratios are listed in **Table 5**. Applied markers allow to distinguish between pyrogenic and petrogenic
as well as traffic and non-traffic sources of PAHs, namely: ANT/(ANT+PHE), FLT/(FLT+PYR), BaA/(BaA+CHR),
IPY/(IPY+BPE), CombPAH/15PAH and BaP/BPE. Calculated ratios for samples taken from residential and
industrial exhibited numbers that point to a domination of pyrogenically formed PAHs. The cross-plots of the PAH
ratios is depicted in **Fig. 5**
**Fig. 5** PAH source apportionment

287       Several markers are indicative of certain combustion sources of PAHs, appointing to gasoline, diesel, crude
oil or grass, coal and wood combustion origins, namely: FLT/(FLT+PYR), BaA/(BaA+CHR), IPY/(IPY+BPE) and
BaP/BPE. The calculated FLT/(FLT+PYR) (0.49-0.51), IPY/(IPY+BPE) (0.30-40) and BaP/BPE (1.20-1.64) values
point to a domination of gasoline, diesel and oil combustion. However, obtained values of FLT/(FLT+PYR) and
BaA/(BaA+CHR) ratios suggested that coal and wood combustion have a certain role in PAHs origination as well. It
is important to note that the shift of heavy and low molecular PAHs ratio towards the heavy ones cannot be
explained only by anthropogenic factor, the degradation of lighter PAHs due to environmental factors such as
photolysis under the direct sun rays in the topsoil layers, as well as thermal degradation, biological uptake and
biodegradation may play a significant role as well (Behymer and Hites 1985; Wild and Jones 1995; Wang 1998;
Johnsen 2005; Choi et al. 2010). These processes are predetermined by physical and chemical properties of the
lighter fraction PAHs such as low molecular weight, high vapor pressure and high volatility rate (Mackay and
Hickie 2000). Volatilization was proved to play the most significant role in the global degradation of the 2- and 3-





ringed PAHs especially. Park et al. (1990) reported that approximately 30% loss of Naphthalene accounts for
volatilization, while for the remaining compounds this process was insignificant. Heavy weight PAHs, i.e. 4-6-ring
compounds, have low solubility in water and low volatility, strong affinity to particulates (BC and SOM, fine
fractions), are less accessible for biological uptake and degradation and thus are more persistent in the environment
(Johnsen 2005; Haritash 2009). It has been proven that PAHs may form nonextractable [$^{14}$C]PAH residues in soil
under the stimulation of microbial activity, which obviously leads to unexpectable lower results while analyzing the
concentrations of Naphthalene, Anthracene, Pyrene and Benzo(a)pyrene in soil samples (Eschenbach et al. 1994;
Eschenbach et al. 1998).
Obtained probabilities for One-way ANOVA revealed no statistically significant differences of total PAH
concentrations in soils between different land uses ($P < 0.05$). Plot of LS Means is depicted in **Fig. 6.**
**Fig. 6.** LS Means plot, differences of PAH levels in soil between land uses.
The differences in levels of individual PAH compounds were tested using Post-hoc Fisher LSD test. The
results showed significant differences of FLT, PYR and PHE concentrations between parkland, residential and
industrial areas ($p$ = 0.01-0.03). The tested hypothesis suggested that PAH levels in urban soil may differ between
areas with different land utilization type, following the order: industrial, residential, parkland. Thus the results of the
study did not prove the tested hypothesis, suggesting the argument of equal PAHs load on the urban soils. The land
use factor is expressed only in distribution of the dominant individual PAHs, particularly FLT and PYR. These
compounds are known to be a part of the PAHs mixtures isolated from the exhaust gases and industrial emissions
(Fernandes et al. 1997; Rehwagen et al. 2005). So not too surprising, that elevated levels of these pollutants are
expected primarily in industrial and transport areas along with surroundings, where maximum input of black carbon
from air pollution sources is noted. PHE representing low molecular weight PAH is a thermodynamically stable tri-
aromatic compound arising from petroleum-hydrocarbon-based releases. Distribution of this contaminant follows
the scheme of potential sources of contamination with petroleum products allocation (**Fig 7**).
**Fig. 7.** Scale of potential sources of contamination with petroleum products (units per square km) with PHE
distribution plots

### 3.3 Risk evaluation of PAHs in soils

Health risk associated with soil contamination with PAHs was assessed using benzo(a)pyrene total potency
equivalents approach (BaP$_{eq}$). The BaP$_{eq}$ for a soil sample is simply calculated by multiplying the concentration of
each PAH in the sample by its benzo(a)pyrene toxic equivalency factor (TEF), given in **Table 6**.
The calculated BaP$_{eq}$ on the average concentration of 15PAH (here and after referred to BaP$_{eq}$-15PAH)
varied between 0.44 to 0.66 mg·kg$^{-1}$ dry soil. The highest BaP$_{eq}$-15PAH mean concentrations were found in
residential and industrial areas – 0.66 and 0.55 mg·kg$^{-1}$ respectively. Parkland areas are characterized by the lower
but still considerable levels of BaP$_{eq}$-15PAH (mean 0.44 mg·kg$^{-1}$). It is to be noted that one single sample taken
from the Kirovskij parkland exhibited a total BaP$_{eq}$ concentration of 1.84 mg·kg$^{-1}$ (The park of 9th January), which
evidently shows that parkland land uses are subjected under a high load of PAHs as well as other land uses.
Obtained values are several times higher than reported total PAHs carcinogenic potencies in a number of studies
(BaP$_{eq}$ of total PAHs): 0.02 mg·kg$^{-1}$ in soils of Viseu and 0.23 mg·kg$^{-1}$ in Lisbon, Portugal (Cachada et al. 2012);
Nadal et al. (2004) reported BaP$_{eq}$ concentrations varying between 0.02 to 0.12 mg·kg$^{-1}$ in soils of Tarragona
County, Spain; 0.18 mg·kg$^{-1}$ in soils of Beijing and 0.24 mg·kg$^{-1}$ in Shanghai, China (Liu et al. 2010; Wang et al.
338 2013).

Finally, obtained BaP total potency equivalents of PAHs were compared with Soil Quality Guideline values
for the direct contact with contaminated soil in respect to particular land use (CCME 2010), setting out the
acceptable level of incremental lifetime cancer risk (ILCR) of $1 \times 10^{-6}$ for BaP$_{eq}$ concentration in soil above the 0.6
to 5.3 mg·kg$^{-1}$ (for each land use). The reported BaP$_{eq}$ for mean total PAH concentrations were above the safe level
of 0.6 mg·kg$^{-1}$. Exposure to these soils through direct contact probably poses a significant risk to human health from
carcinogenic effects of PAHs, even in urban parklands.

### 4. Conclusions

Results of the study demonstrated that soils within urban areas with different land utilization types are
characterized by common loads of PAHs generally attributed to high traffic density of the city. Considerable levels
of soil contamination with PAHs were noted. The common tendency in PAH distribution patterns between
investigated sites clearly indicates the common source of PAHs in urban soils. A larger portion of high molecular
weight PAHs along with determined molecular ratios suggest the predominance of pyrogenic sources, mainly




attributed to combustion of gasoline, diesel and oil. Petrogenic sources of PAHs have a significant portion as well defining the predominance of petroleum associated low molecular weight PAHs such as phenanthrene. Derived concentrations of 7 carcinogenic PAHs as well as calculated BaP total potency equivalents were multiple times higher than reported in a number of other studies, indicating a significant risk for human health in case of direct contact. One-way ANOVA results showed significant differences in levels of pyrene, fluoranthene and phenanthrene – the most abundant individual PAHs in examined sampled, between parkland, residential and industrial land uses, suggesting the influence of land use factor on distribution of these pollutants. Further study with an application of complex statistical methods is needed such as principal component analysis which would contribute to precision of PAHs sources allocation.

**Acknowledgments**

This work was supported by Russian Foundation for Basic Research, project № 15-34-20844

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




**Table 1.** Description of the study area

| Characteristics | Units | Primorsky District | Vasileostrovsky District | Kirovsky District |
|---|---|---|---|---|
| S | km$^2$ | 109.87 | 17.1 | 47.10 |
| Population | - | 534 646 | 211 048 | 334746 |
| Industries | units | 250 | 350 | 70 |
| Number of potential contamination sources with petroleum products | units | 14 | 7 | 10 |
| Density of potential contamination sources | units per km$^2$ | 0.13 | 0.41 | 0.21 |
| CH$_x$ emissions from stationary sources in 2014 | thousand tons | 0.556 | 0.034 | 0.708 |
| VOC emissions from stationary sources in 2014 | thousand tons | 0.153 | 0.099 | 0.545 |
| BC emissions from stationary sources in 2014 | thousand tons | 0.237 | 0.037 | 0.174 |

**Table 2.** Physicochemical properties of the studied soils

| District | Land use | Soil type. FAO | TOC | Clay |
|---|---|---|---|---|
| | | | % | |
| Primorskij | Parkland | Mollic Technosol | 4.10 ± 0.01 | 5.83 ± 0.21 |
| | Residential | Urbic Technosol | 3.82 ± 0.03 | 7.43 ± 0.06 |
| | Industrial | Urbic Technosol | 5.49 ± 0.02 | 8.50 ± 0.10 |
| Vasileostrovskij | Parkland | Mollic Technosol | 5.39 ± 0.01 | 7.3 ± 0.20 |
| | Residential | Urbic Technosol | 6.41 ± 0.02 | 1.87 ± 0.12 |
| | Industrial | Urbic Technosol | 5.28 ± 0.02 | 3.27 ± 0.15 |
| Kirovskij | Parkland | Mollic Technosol | 4.19 ± 0.03 | 7.5 ± 0.10 |
| | Residential | Urbic Technosol | 4.80 ± 0.03 | 3.27 ± 0.15 |
| | Industrial | Urbic Technosol | 3.09 ± 0.02 | 7.67 ± 0.06 |



**Table 3.** PAH mean concentrations in soils of St. Petersburg (mg·kg$^{-1}$).

| Compound | Parkland (n = 9) | | | Residential (n = 9) | | | Industrial (n = 9) | | | P One-way ANOVA ($\alpha = 0.05$) |
|---|---|---|---|---|---|---|---|---|---|---|
| | Mean ± SD | Max | Min | Mean ± SD | Max | Min | Mean ± SD | Max | Min | |
| NAP | 0.06 ± 0.08 | 0.28 | 0.03 | 0.05 ± 0.02 | 0.07 | 0.00 | 0.09 ± 0.07 | 0.21 | 0.00 | 0.42 |
| ANA | 0.02 ± 0.06 | 0.18 | 0.00 | 0.00 | 0.01 | 0.00 | 0.00 | 0.03 | 0.00 | 0.41 |
| FLU | 0.10 ± 0.06 | 0.23 | 0.05 | 0.17 ± 0.11 | 0.40 | 0.03 | 0.17 ± 0.11 | 0.31 | 0.06 | 0.28 |
| PHE | 0.16 ± 0.13 | 0.45 | 0.05 | 0.26 ± 0.17 | 0.47 | 0.03 | 0.36 ± 0.22 | 0.65 | 0.07 | 0.10 |
| ANT | 0.06 ± 0.11 | 0.37 | 0.01 | 0.04 ± 0.04 | 0.11 | 0.00 | 0.05 ± 0.03 | 0.09 | 0.01 | 0.87 |
| FLT | 0.18 ± 0.07 | 0.35 | 0.09 | 0.69 ± 0.52 | 1.49 | 0.04 | 0.72 ± 0.48 | 1.50 | 0.11 | 0.02 |
| PYR | 0.18 ± 0.08 | 0.35 | 0.09 | 0.74 ± 0.55 | 1.67 | 0.04 | 0.70 ± 0.46 | 1.50 | 0.16 | 0.02 |
| BaA | 0.19 ± 0.17 | 0.53 | 0.04 | 0.35 ± 0.26 | 0.64 | 0.02 | 0.30 ± 0.20 | 0.67 | 0.07 | 0.32 |
| CHR | 0.15 ± 0.14 | 0.44 | 0.01 | 0.31 ± 0.24 | 0.69 | 0.02 | 0.28 ± 0.18 | 0.54 | 0.07 | 0.24 |
| BbF | 0.23 ± 0.21 | 0.69 | 0.05 | 0.46 ± 0.30 | 0.84 | 0.02 | 0.41 ± 0.30 | 1.00 | 0.10 | 0.21 |
| BkF | 0.15 ± 0.17 | 0.56 | 0.02 | 0.19 ± 0.14 | 0.36 | 0.01 | 0.16 ± 0.11 | 0.33 | 0.04 | 0.82 |
| BaP | 0.22 ± 0.22 | 0.70 | 0.04 | 0.43 ± 0.32 | 0.87 | 0.02 | 0.34 ± 0.23 | 0.73 | 0.07 | 0.30 |
| DBA | 0.03 ± 0.06 | 0.18 | 0.00 | 0.02 ± 0.01 | 0.04 | 0.00 | 0.02 ± 0.03 | 0.08 | 0.00 | 0.93 |
| BPE | 0.17 ± 0.14 | 0.46 | 0.04 | 0.29 ± 0.21 | 0.52 | 0.01 | 0.27 ± 0.20 | 0.69 | 0.06 | 0.39 |
| IPY | 0.12 ± 0.15 | 0.49 | 0.00 | 0.17 ± 0.17 | 0.45 | 0.01 | 0.15 ± 0.13 | 0.38 | 0.00 | 0.76 |
| ∑15PAH | 2.02 ± 1.50 | 4.78 | 0.58 | 4.17 ± 2.91 | 8.10 | 0.33 | 4.02 ± 2.61 | 8.06 | 0.86 | 0.16 |
| ∑7PAH$^a$ | 1.08 ± 1.04 | 3.18 | 0.21 | 1.94 ± 1.36 | 3.47 | 0.10 | 1.66 ± 1.13 | 3.20 | 0.36 | 0.35 |

NAP – naphthalene,ANA – acenaphthene,FLU – fluorene,PHE – phenanthrene,ANT – anthracene,FLT – fluoranthene,PYR – pyrene,BaA – benzo(a)anthracene,CHR – chrysene,BbF – benzo(b)fluoranthene,BkF – benzo(k)fluoranthene,BaP - benzo(a)pyrene,DBA – dibenz(ah)anthracene,BPE – benzo(g,h,i)perylene,IPY – indeno(1,2,3-cd) pyrene.

$^a$ Carcinogenic PAHs: chrysene,benzo(a)anthracene,benzo(b)fluoranthene,benzo(k)fluoranthene,benzo(a)pyrene,indeno(1,2,3-cd) pyrene and dibenz(ah)anthracene.



**Table 4.** Reported total concentrations of PAHs in urban soils (mg·kg$^{-1}$ dry weight) from a number of studies

| Location | Study area | Concentrations (mg·kg$^{-1}$ d.w.) | ∑PAH | Reference |
|---|---|---|---|---|
| Huston, TX, USA | Urban / suburban | 0.2-2.2 | 23 | Hwang et al. (2002) |
| Mexico City, Mexico | Urban / industrial | 0.20-1.10 | 17 | Hwang et al. (2003) |
| Beijing, China | Urban | 0.22-27.82 | 16 | Tang et al. (2005) |
| New Orlean, USA | Urban | 3.73 (median) | 16 | Mielke et al. (2001) |
| Tarragona County, Catalonia, Spain | Urban / residential / industrial | 0.11-1.00 | 16 | Nadal et al. (2004) |
| Swiss soil monitoring system (NABO), Switzerland | Urban parkland / semiurban | 0.05-0.62 | 16 | Bucheli et al. (2004) |
| Tallinn, Estonia | Urban | 2.20±1.40 | 12 | Trapido (1999) |
| Linz, Austria | Industrial | 1.45 (median) | 18 | Weiss et al. (1994) |
| Tokushima, Japan | Urban | 0.61 | 13 | Yang et al. (2002) |
| Shanghai, China | Main urban | 0.13-8.65 / 0.08-7.22 | 26 / 16 | Wang et al. (2013) |
| El-Tabbin, Egypt | Urban / industrial | 0.05-5.56 | 16 | Havelcová et al. (2014) |
| Phoenix, Arizona, USA | Urban (highways) | 0.06-10.12 | 20 | Marusenko et al. (2010) |

**Table 5.** PAH ratios in studied soils

| Ratio | Parkland | Indicated source (origin) | Residential | Indicated source (origin) | Industrial | Indicated source (origin) |
|---|---|---|---|---|---|---|
| ANT / (ANT + PHE) | 0.19 | Pyrogenic | 0.09 | Petrogenic | 0.12 | Pyrogenic |
| FLT / (FLT + PYR) | 0.51 | Grass. coal and wood combustion | 0.49 | Gasoline. diesel and crude oil combustion | 0.50 | Gasoline. diesel and crude oil combustion |
| BaA / (BaA + CHR) | 0.58 | Grass. coal and wood combustion | 0.52 | Grass. coal and wood combustion | 0.51 | Grass. coal and wood combustion |
| IPY / (IPY + BPE) | 0.30 | Liquid fossil fuel combustion | 0.40 | Liquid fossil fuel combustion | 0.34 | Liquid fossil fuel combustion |
| BaP / BPE | 1.20 | Traffic sources | 1.64 | Traffic sources | 1.31 | Traffic sources |
| CombPAH / ∑PAH | 0.79 | Combustion dominated source | 0.80 | Combustion dominated source | 0.81 | Combustion dominated source |





**Table 6.** PAH concentrations in urban soils, expressed in BaP$_{eq}$, mg·kg$^{-1}$

| Compound | Parkland | | | Residential | | | Industrial | | | TEF[a] |
|---|---|---|---|---|---|---|---|---|---|---|
| | Mean × TEF | Max × TEF | Min × TEF | Mean × TEF | Max × TEF | Min × TEF | Mean × TEF | Max × TEF | Min × TEF | |
| NAP | 0.00006 | 0.00028 | 0.00003 | 0.00005 | 0.00007 | 0.00 | 0.00009 | 0.00021 | 0.00 | 0.001 |
| ANA | 0.0002 | 0.00018 | 0.00 | 0.00 | 0.00001 | 0.00 | 0.00 | 0.00003 | 0.00 | 0.001 |
| FLU | 0.0001 | 0.00023 | 0.00005 | 0.00017 | 0.0004 | 0.00003 | 0.00017 | 0.00031 | 0.00006 | 0.001 |
| PHE | 0.00016 | 0.00045 | 0.00005 | 0.00026 | 0.00047 | 0.00003 | 0.00036 | 0.00065 | 0.00007 | 0.001 |
| ANT | 0.0006 | 0.0037 | 0.0001 | 0.0004 | 0.0011 | 0.00 | 0.0005 | 0.0009 | 0.0001 | 0.01 |
| FLT | 0.00018 | 0.00035 | 0.00009 | 0.00069 | 0.00149 | 0.00004 | 0.00072 | 0.0015 | 0.00011 | 0.001 |
| PYR | 0.00018 | 0.00035 | 0.00009 | 0.00074 | 0.00167 | 0.00004 | 0.0007 | 0.0015 | 0.00016 | 0.001 |
| BaA | 0.019 | 0.053 | 0.004 | 0.035 | 0.064 | 0.002 | 0.03 | 0.067 | 0.007 | 0.10 |
| CHR | 0.0015 | 0.0044 | 0.0001 | 0.0031 | 0.0069 | 0.0002 | 0.0028 | 0.0054 | 0.0007 | 0.01 |
| BbF | 0.023 | 0.069 | 0.005 | 0.046 | 0.084 | 0.002 | 0.041 | 0.10 | 0.01 | 0.10 |
| BkF | 0.015 | 0.0560 | 0.002 | 0.019 | 0.036 | 0.001 | 0.016 | 0.033 | 0.004 | 0.10 |
| BaP | 0.22 | 0.7 | 0.04 | 0.43 | 0.87 | 0.02 | 0.34 | 0.73 | 0.07 | 1.00 |
| DBA | 0.15 | 0.90 | 0.00 | 0.10 | 0.20 | 0.00 | 0.10 | 0.40 | 0.00 | 5.00 |
| BPE | 0.0017 | 0.0046 | 0.0004 | 0.0029 | 0.0052 | 0.0001 | 0.0027 | 0.0069 | 0.0006 | 0.01 |
| IPY | 0.012 | 0.049 | 0.00 | 0.017 | 0.045 | 0.001 | 0.015 | 0.038 | 0.00 | 0.10 |
| ∑15PAH | 0.4435 | 1.84154 | 0.05191 | 0.65531 | 1.31631 | 0.02644 | 0.55004 | 1.3854 | 0.0928 | |
| ∑7PAH[a] | 0.4405 | 1.8314 | 0.0511 | 0.6501 | 1.3059 | 0.0262 | 0.5448 | 1.3734 | 0.0917 | |

[a]Values of the Toxic equivalency factors proposed by Nisbet and Lagoy (1992).

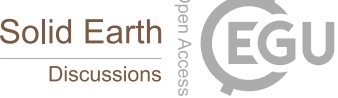

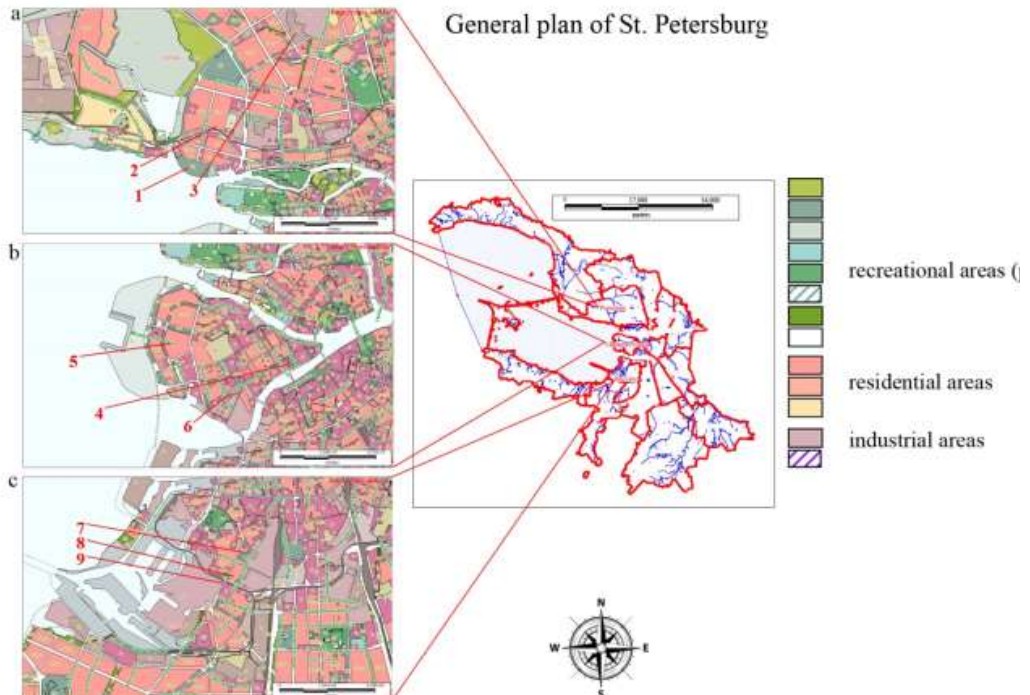

**Fig. 1.** Location of the soil sampling sites.

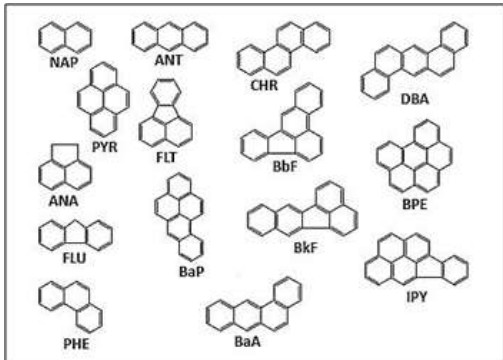

**Fig, 2.** Structures of the studied PAH compounds.



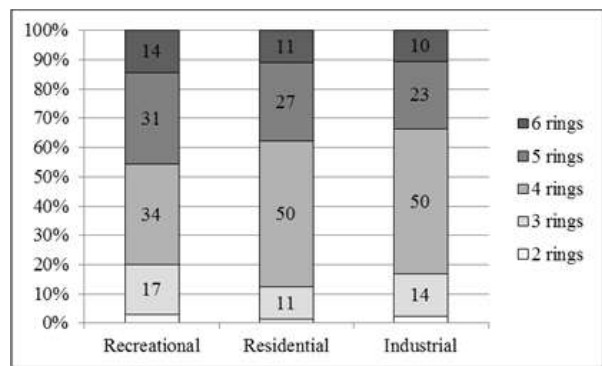

**Fig 3.** Distribution pattern of PAHs with different ring numbers in studied soils

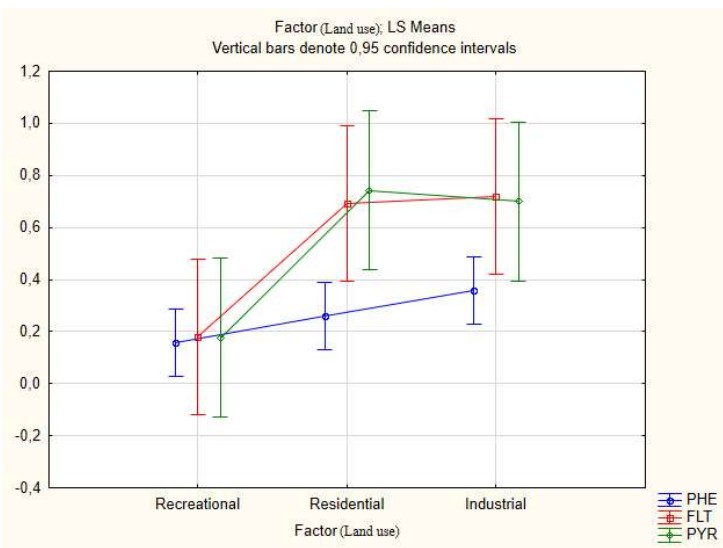

**Fig 4.** Composition of PAH mixtures in studied soils


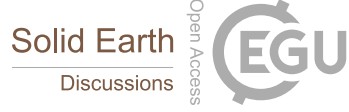

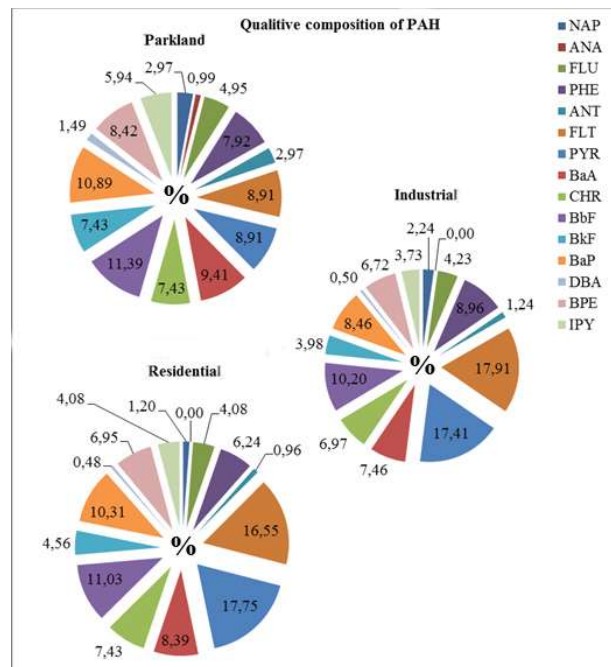

**Fig. 5** PAH source apportionment



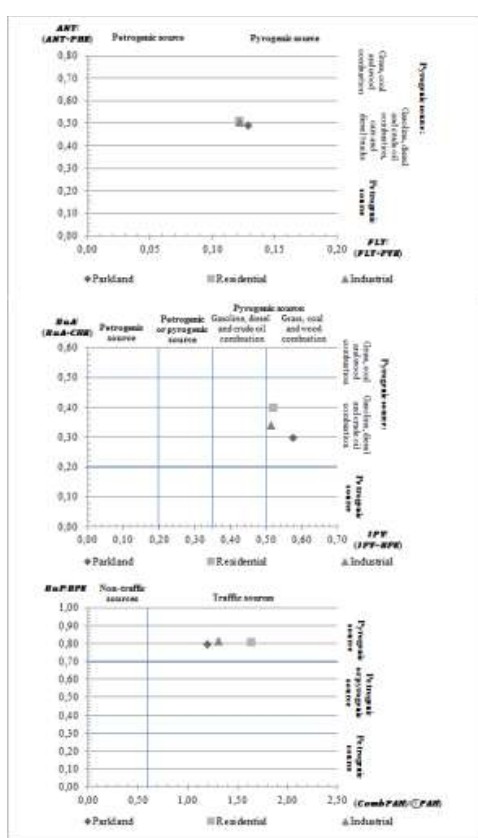

**Fig. 6.** LS Means plot, differences of PAH levels in soil between land uses

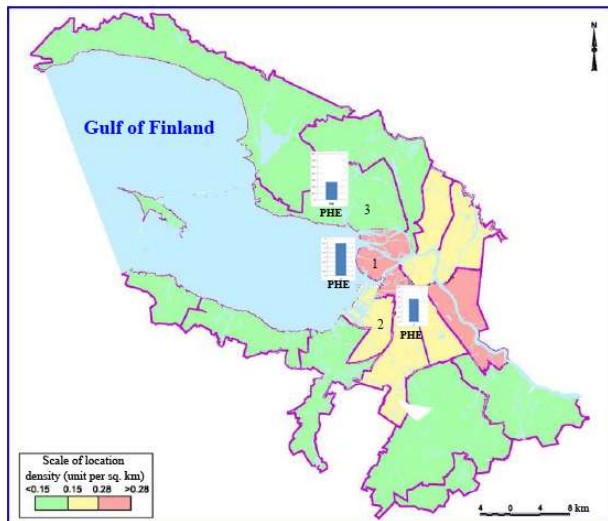

**Fig. 7.** Scale of potential sources of contamination with petroleum products (units per square km) with PHE distribution plots