# Peer review of "POLYCYCLIC AROMATIC HYDROCARBONS IN THE SURFACE SOILS OF ST. PETERSBURG Shamilishvily GA1, Abakumov EV1, Gabov DN2"

_Solid Earth, 2017_

## Short Comment (SC1) · 7 Sep 2017

This paper is interesting and urgent in sense of soil and human health in large cities. The authors have investigated a qualitative and quantitative composition of 15 priority PAHs in urban soils on the example of Saint-Petersburg (Russian Federation). I found this paper of good quality and ready to publish. Several comments: I suggest include more parameters in table 2 to characterize studied soils more precisely - some general properties such as: soil color in accordance with the Munsell Color System, pH, nitrogen content. The quality of diagrams at the figure 6 is very poor. Nothing is clear at the

picture. Please, change the picture or provide some discussions of p-values from the table 3, but not from figure 6. Please provide the short review of natural soil diversity in Saint-Petersburg and Leningrad region, close to the study plots, with references (in Materials and Methods).

---

## Short Comment (SC2) · 10 Sep 2017

PAHs undoubtedly pose a threat to living organisms. In an urban environment, which is problematic due to many related factors and ease way of migration, their quantity should be closely monitored. The results of the study confirm the need to monitor sources of pollution and the need of reduction sources of emissions. The resolution of Fig. 1. could be better, Fig. 2 is unnecessary. Fig. 6 is blurred.

---

## Referee Comment (RC1) · Anonymous Referee #1 · 1 Oct 2017

All manuscript

a) The language is partly insufficient. It should be helpful to get the revision of a native English speaking person. b) the abstract is too long. c) there are not related literatures in many expressions. d) Too many references (around 75). Some references are cited only once and they can be replaced by others.

Specific comments

Line 33-35: you should list the related literature(s) for your expression. Line 35-37:

[Figure]

in my opinion, not all PAHs should be classified as POPs, such as naphthalene. Line 66: insert white space in 'ofreports'. Line 66-67: what is the basis of 'thousands of reports....' Line 81: km2 Line 276: Regarding to the methods of PAH sources, you could refer the literature by Wang (2017, pedosphere) or Wang (2015, Sci Total Environ)

---

## Referee Comment (RC2) · Anonymous Referee #2 · 9 Oct 2017

All manuscript: This study investigates 15 PAHs in urban soils from different land uses. The authors show us new data, however, it seems like a report in current version. Authors need to improve their presentation and discuss further. For example, the mechanisms of PAHs accumulation in soils, different pattern of PAHs among different land uses and the source analysis, etc. Specific comments: 1. Line91-94: It is not the aim of this study. It is what author did in this study. 2. Don't only leave the figure caption in text. 3. Line147: Figure 2 is not necessary. 4. Why measured 15 PAHs not 16 PAHs? 5. Check figures. 6. Line 237-246: Figure3 and Figure 4 are repeated. And

this part should be moved to source analysis. 7. Line266-275: Authors don't need to state the details of other studies in text due to they are showing in Table 4. 8. Figure 5 and Table 5 are repeated. The resolution of figure 5 is not enough. 9. Line 287-306: Authors state the weakness of diagnostics ratios. So, did author have confidence of the conclusion deduced by this method? 10. Line 310-321: Authors suggest no significant difference in total PAHs among different land uses, which is not consistent with many other studies. I can understand there is no difference in total PAHs. However, the authors should discuss more about the difference in PAH compositions or PAH patterns among different land uses. 11. I don't think it is appropriate to draw figure 7 based on 9 sites.

---

## Author Comment (AC1) · 9 Oct 2017

Thank you very much, we have took into account all your suggestions, prticularly, we have added data on the soil pH, nitrogen content and mansell colour chart indices to the table 2, added short introduction about natural soils of St. Peterburg region and clarified fig. 5 (there was some confusion with figure numeration, now we have put figures in correct order).

---

## Author Comment (AC2) · 9 Oct 2017

Thank you very much, we have taken into account your suggestons, in particular we have deleted fig. 2 as unnesesery, clarified the resolution of fig. 5, unfortunetely we cant clarify resolution of the pic number 1, because this pic. was obtained in a low resolution JPEG format through the online application.

---

## Referee Comment (RC3) · Y.-G. Gu (Referee) · 10 Oct 2017

Y.-G. Gu (Referee)

hydrobio@163.com

I have carefully read through the MS titled "Polycyclic aromatic hydrocarbon in urban soils of the Eastern European megalopolis: distribution, 1 source identification and cancer risk evaluation". I evaluated the originality, significance and technical quality of the work and the length and clarity of this manuscript. As we know, Residents are primarily exposed to PAHs via three main pathways: ingestion; dermal absorption; and inhalation. According to your manuscript, the risk assessment is an important part, but this MS lack some key information related to risk assessment. Please provide

each pathway risk assessment for children and adults. And some references may be help you and should be cited in your manuscript as below: Archives of Environmental Contamination and Toxicology, 2017, 72(4): 496-504

---

## Author Comment (AC3) · 31 Oct 2017

a) The language is partly insufficient. It should be helpful to get the revision of a native English speaking person. - We will b) The abstract is too long. - Abstract was shortened according to suggestion. c) there are not related literatures in many expressions. d) Too many references (around 75). Some references are cited only once and they can be replaced by others.

- Reference list was shortened, analogous references were removed

[Figure]

Specific comments

Line 33-35: you should list the related literature(s) for your expression.

- checked it.

Line 35-37

in my opinion, not all PAHs should be classified as POPs, such as naphthalene. - completely agree with you, we rephraised the statement and made it be more accurate.

Line 66: insert white space in 'ofreports'. - Done;

Line 66-67: what is the basis of 'thousands of reports. . ..' - Related references were added;

Line 81: km2 - put 2 to upper index;

Line 276: Regarding to the methods of PAH sources, you could refer the literature by Wang (2017, pedosphere) or Wang (2015, Sci Total Environ);

- Thank you, we found these referenses very useful. Added it to the list.

–––––––––––––––––––––––––––

---

## Author Comment (AC4) · 31 Oct 2017

All manuscript: This study investigates 15 PAHs in urban soils from different land uses. The authors show us new data, however, it seems like a report in current version. Authors need to improve their presentation and discuss further. For example, the mechanisms of PAHs accumulation in soils, different pattern of PAHs among different land uses and the source analysis, etc. Specific comments: 1. Line 91-94: It is not the aim of this study. It is what author did in this study. - Yes, it were objectives of our study, the aim wa to aimed to test the hypothesis on the PAH load differences between urban

territories with different land use scenarios. 2. Don't only leave the figure caption in text.

3. Line147: Figure 2 is not necessary. - Deleted according to suggestions.

4. Why measured 15 PAHs not 16 PAHs?

- Due to technical reasons, we did not have soil reference materials containing acenaphthylene for QC.

5. Check figures.

- Checked all the figures, numeration was corrected as well as figure references in the text.

6. Line 237-246: Figure3 and Figure 4 are repeated. And this part should be moved to source analysis.

- Corrected according to suggestions.

7. Line266-275: Authors don't need to state the details of other studies in text due to they are showing in Table 4.

- Details were removed from the text according to suggestions.

8. Figure 5 and Table 5 are repeated. The resolution of figure 5 is not enough.

- Checked, resolution was improoved.

9. Line 287-306:

Authors state the weakness of diagnostics ratios. So, did author have confidence of the conclusion deduced by this method? We are going to conduct PCA once we will get more data.

10. Line 310-321: Authors suggest no significant difference in total PAHs among different land uses, which is not consistent with many other studies. I can understand there

is no difference in total PAHs. However, the authors should discuss more about the difference in PAH compositions or PAH patterns among different land uses.

- Actually, it is stated that there are no significant differences in total PAHs concentartions as well as in composition of PAH mixuters only between residential and indutrial areas, since they are very loaded, in contrast to recreational area.

―――――――――――――――――――――

---

## Author Comment (AC5) · 31 Oct 2017

I have carefully read through the MS titled "Polycyclic aromatic hydrocarbon in urban soils of the Eastern European megalopolis: distribution, 1 source identification and cancer risk evaluation". I evaluated the originality, significance and technical quality of the work and the length and clarity of this manuscript. As we know, Residents are primarily exposed to PAHs via three main pathways: ingestion; dermal absorption; and inhalation. According to your manuscript, the risk assessment is an important part, but this MS lack some key information related to risk assessment. Please provide

each pathway risk assessment for children and adults. And some references may be help you and should be cited in your manuscript as below: Archives of Environmental Contamination and Toxicology, 2017, 72(4): 496-504

- We have evluated cancer risk related to three routes of exposure for both children and adults using a RAIS Risk Exposure Models for Chemicals. Thank you for suggestion.

---

## Author Comment (AC7) · 31 Oct 2017

All changes are marked yellow

[revised manuscript text omitted]

---

## Author Comment (AC8) · 31 Oct 2017

All changes are marked yellow
* * *
[Figure]

[revised manuscript text omitted]

---

## Author Comment (AC9) · 31 Oct 2017

All changes are marked yellow
* * *
[Figure]

**Polycyclic aromatic hydrocarbon in urban soils of the Eastern European megalopolis: distribution,**
**identification and cancer risk evaluation**

[revised manuscript text omitted]

---

## Author Comment (AC10) · 31 Oct 2017

All changes are marked yellow
* * *
[Figure]

[revised manuscript text omitted]

---

## Author Comment (AC11) · 31 Oct 2017

All changes are marked yellow

[revised manuscript text omitted]

---

## Referee Comment (RC4) · Anonymous Referee #4 · 2 Nov 2017

The manuscript is devoted to the actual topic of polycyclic aromatic hydrocarbons (PAH) pollution of urban soils in cities. This is up to date research compelling the lack of data on PAH distribution and cancer risk evaluation in Saint-Petersburg – one of the largest cities in the Eastern Europe. I would suggest this manuscript to be published after the major revision. 1. The key result of the manuscript is quite questionable – "Total PAH concentrations . . .showed no significant differences between land utilization types" (Line 18-19). A lot of published researches prove the opposite finding – clear differentiation between zones (parkland, residential, industrial) exists. To my mind the

absence of differentiation may be driven by the specific of sampling procedure – may be the soil samples were excavated in the vicinity of highways/roads in all the zones. The procedure of sampling is not clearly described by the authors – the dense of road system and the distance from the roads of every sampling plots should be specified as must for every zone.

Please, provide the detailed scheme/map of sampling sites putting sampling plots on it. Highways/roads location (the distance from the roads to sampling plots, the distance between sampling plots) should be clear, production plants location and dominating wind directions should be specified.

Fig . 1 is not informative and too small to realize the location of sampling plots in road and production plants system.

2. Poor characteristics of soil sampling sites and absence of information on soil sampling plots – their location specific (distance from roads as mentioned above), traffic intensity of closest roads, dominating wind direction, vegetation type, relief and landscape specific, population density (Line 105 and further).

3. Sampling strategy and procedure (Line 120) is not clear and should be rewritten: 3.1. Specify quantity of sampling plots 3.2. Specify distance between sampling plots within sampling site and between them 3.3. Was the sampling depth different at sites – what does mean the phrase "Soil depth selected for sampling. . ." (Line 128-130) 3.4. What is behind the phrase "Sampling pattern. . ." (Line 130-132) and "This technique enables. . ." (Line 140-141). 3.5. Specify the weight/volume of one "initial" soil sample excavated within sampling plot before mixing (average sample formation) 3.6. Was the quantity of samples within all the sampling plots the same 3.7. As I understood it was 3 different sampling plots per functional zone in sampling site. Why the GPS location is only one per zone in Fig 1-description?

4. Lines 188-198 – should be moved to Results and discussion section

5. Line 220-221 is in conflict with Line 17-19 ("Total PAH"). Please, explain.

6. The structuring of the manuscript should be improved to make it more logical and clear in line with key objective/aim – to test the hypothesis on the PAH loading differences between urban territories of different use scenarios (functional zones). I would suggest to structure all the sections in Results and discussion part in the common way: 1 – key findings prior to different zones, 2 – discussion 7. Line 276 "Determination of the PAH sources and statistics". Why "statistics" is highlighted in this section? Statistics relates to all the sections – does not it?

8. Conclusions should be revised prior the above comments

9. Technical remarks: 10.1 Line 1: hydrocarbons instead of hydrocarbon 10.2 Line 123-124: no noun to "were combined" 10.3 Fig.2 – no need, it is general knowledge 10.4 Fig. 1-description: Primorskiy and other names instead of Primorskij

---

## Author Comment (AC12) · 22 Dec 2017

Dear reviewer, thank you for your labor and patience, we have tried to respond to all your comments: ————————————————————————————————————————————————————————- The manuscript is devoted to the actual topic of polycyclic aromatic hydrocarbons (PAH) pollution of urban soils in cities. This is up to date research compelling the lack of data on PAH distribution and cancer risk evaluation in Saint-Petersburg – one of the largest cities in the Eastern Europe. I would suggest this manuscript to be published after the major revision. 1. The key result of the manuscript

is quite questionable – "Total PAH concentrations . . .showed no significant differences between land utilization types" (Line 18-19). A lot of published researches prove the opposite finding – clear differentiation between zones (parkland, residential, industrial) exists. To my mind the absence of differentiation may be driven by the specific of sampling procedure – may be the soil samples were excavated in the vicinity of highways/roads in all the zones. The procedure of sampling is not clearly described by the authors – the dense of road system and the distance from the roads of every sampling plots should be specified as must for every zone. 1. Please, provide the detailed scheme/map of sampling sites putting sampling plots on it. Highways/roads location (the distance from the roads to sampling plots, the distance between sampling plots) should be clear, production plants location and dominating wind directions should be specified. Fig . 1 is not informative and too small to realize the location of sampling plots in road and production plants system. - We have redesigned figure 1, made it even bigger and understandable, divided it to fig 1 a, b, c parts showing each study area individually with sampling plots, roads and production plants location clearly marked.

2. Poor characteristics of soil sampling sites and absence of information on soil sampling plots – their location specific (distance from roads as mentioned above), traffic intensity of closest roads, dominating wind direction, vegetation type, relief and landscape specific, population density (Line 105 and further).

- We have added detailed information on the sampling plots to the supplementary materials, now it looks like 3 big tablets.

3. Sampling strategy and procedure (Line 120) is not clear and should be rewritten:

3.1. Specify quantity of sampling plots - added missing information according to suggestion;

3.2. Specify distance between sampling plots within sampling site and between them - added missing information according to suggestion;
3.3. Was the sampling depth different at sites - it was common 0-20 cm, we specified it in the text;

– what does mean the phrase "Soil depth selected for sampling. . ." (Line 128-130) - we rephrased this sentence to: "Depth of sampling is a function of exposure routes..."

3.4.What is behind the phrase "Sampling pattern. . ." (Line 130-132) and "This technique enables. . ." (Line 140-141). - Replaced "Sampling pattern. . ." with "Sampling scheme" and "This technique enables. . ." with "This method allows to prevent".

3.5. Specify the weight/volume of one "initial" soil sample excavated within sampling plot before mixing (average sample formation) - added missing information according to suggestion;

3.6. Was the quantity of samples within all the sampling plots the same - nope, it differed.

3.7. As I understood it was 3 different sampling plots per functional zone in sampling site. Why the GPS location is only one per zone in Fig 1-description? - No, the quantity of sampling plots ranged between 2 and 5, we added missing information and specified it in the text.

4. Lines 188-198 – should be moved to Results and discussion section - corrected according to suggestions.

5. Line 220-221 is in conflict with Line 17-19 ("Total PAH"). Please, explain - we are sorry for this mistake, our callegue put old data, the given p values in table 3 and conclusions in the text in this redaction of manuscript was not actual, this was only for Primorskiy district, this conclusion and p values in table 3 are from the previos version of our manuscript, when comparison was made between studied districts nor the land uses, now we have grouped data for each land use type from all the studied districts together to create a bigger dataset for One-Way ANOVA. Differences in total PAHs and 7PAHs tend to be significant from this point of view. We added actual p values in tablet

3 and revised conclusions in the text.

6. The structuring of the manuscript should be improved to make it more logical and clear in line with key objective/aim – to test the hypothesis on the PAH loading differences between urban territories of different use scenarios (functional zones). I would suggest to structure all the sections in Results and discussion part in the common way: 1 – key findings prior to different zones, 2 – discussion - checked

7. Line 276 "Determination of the PAH sources and statistics". Why "statistics" is highlighted in this section? Statistics relates to all the sections – does not it? - it does, divided it to individual section.

8. Conclusions should be revised prior the above comments - checked

9. Technical remarks: 10.1 Line 1: hydrocarbons instead of hydrocarbon - checked

10.2 Line 123-124: no noun to "were combined" - checked

10.3 Fig.2 – no need, it is general knowledge - checked

10.4 Fig. 1-description: Primorskiy and other names instead of Primorskij - checked

Please also note the supplement to this comment:
https://www.solid-earth-discuss.net/se-2017-54/se-2017-54-AC12-supplement.zip